# Rhythmic properties of *Sciaena umbra* calls across space and time in the Mediterranean Sea

**Marta Picciulin**[1], **Marta Bolgan**[2], **Lara S. Burchardt**[3,4] *

1 CNR-National Research Council, ISMAR—Institute of Marine Sciences, Venice, Italy, 2 Ocean Science Consulting Limited, Dunbar, United Kingdom, 3 Max-Planck-Institut for Psycholinguistics, Nijmegen, Netherlands, 4 Leibniz-Zentrum Allgemeine Sprachwissenschaft, Berlin, Germany

* l.s.burchardt@gmx.de

**Data Availability Statement:** Data and codes to reproduce the results are now available at the public Github repository: https://github.com/LSBurchardt/Rhythmic-properties-Sciaena-umbra

## Abstract

In animals, the rhythmical properties of calls are known to be shaped by physical constraints and the necessity of conveying information. As a consequence, investigating rhythmical properties in relation to different environmental conditions can help to shed light on the relationship between environment and species behavior from an evolutionary perspective. *Sciaena umbra* (fam. Sciaenidae) male fish emit reproductive calls characterized by a simple isochronous, i.e., metronome-like rhythm (the so-called R-pattern). Here, *S. umbra* R-pattern rhythm properties were assessed and compared between four different sites located along the Mediterranean basin (Mallorca, Venice, Trieste, Crete); furthermore, for one location, two datasets collected 10 years apart were available. Recording sites differed in habitat types, vessel density and acoustic richness; despite this, *S. umbra* R-calls were isochronous across all locations. A degree of variability was found only when considering the beat frequency, which was temporally stable, but spatially variable, with the beat frequency being faster in one of the sites (Venice). Statistically, the beat frequency was found to be dependent on the season (i.e. month of recording) and potentially influenced by the presence of soniferous competitors and human-generated underwater noise. Overall, the general consistency in the measured rhythmical properties (isochrony and beat frequency) suggests their nature as a fitness-related trait in the context of the *S. umbra* reproductive behavior and calls for further evaluation as a communicative cue.

## Introduction

Acoustic communication plays an important role in mediating intraspecific interactions in terrestrial and aquatic animals across a wide taxonomic spectrum [1]. In many marine animals, including marine mammals, fish, and crustaceans acoustic signals can mediate social interactions during reproduction, resource defense, group cohesion, anti-predator strategies, and individual recognition [2, 3]. A strong selection toward acoustic communication as a behavioral character providing fitness-related advantages has been recently proven across vertebrate

**Funding:** The author(s) received no specific funding for this work.

**Competing interests:** The authors have declared that no competing interests exist.

evolution [4, 5]; in fishes, sound production has evolved independently over 30 times as a potential result of exaptation and potentially occurs in nearly two-thirds of actinopterygian species [5, 6].

In many taxa, biologically relevant information is encoded in species-specific signals and in their temporal organization. Vocal rhythms are known to encode individual or context-related information in terrestrial vertebrates such as primates, birds, and anurans [7–10], and in marine vertebrates such as Phocidae and whales [11–15]. The rhythmical properties of fish vocalizations are less documented, despite qualitative assessments indicating their existence and their potential key role in crucial behaviors [16–18]. A variable rhythm of sound production was found in the Lusitanian toadfish (*Halobatrachus didactylus*) [19]. Furthermore, an inter-specific variability of rhythmical properties was found between the three most common fish sound types in the Mediterranean basin [20], which have been recorded over a wide spatio-temporal range [17, 18, 21–28], i.e. the rochei's snake blenny (*Ophidion rochei*) produces a random temporal succession of sounds, the brown meagre (*Sciaena umbra*) emits isochronous, i.e., metronome-like calls (the so-called regular pattern or R-pattern), while more complex rhythmical properties characterized the so-called 'Kwa' vocalizations attributable to *Scorpaena spp.* [20]. Among these species, *S. umbra*, one of the five Mediterranean Sciaenidae, was the first to be monitored in the Mediterranean basin thanks to Passive Acoustic Monitoring [16, 17, 23, 27, 29–31].

*Sciaena umbra* is a small-sized, sedentary, and gregarious demersal fish inhabiting rocky bottoms and *Posidonia oceanica* beds across the Mediterranean Sea [32–34]. *S. umbra* has been largely impacted by overfishing and is nowadays listed in Annex III (Protected Fauna Species) of the Barcelona Conventions and is classified as a vulnerable fish species by the International Union for Conservation of Nature [35, 36]. Like other Sciaenids [37–41], *S. umbra* emits drumming, multi-pulsed sounds during its reproductive period [16, 42] but only males are vocal given that the sonic apparatus is absent in females [31]. In *S. umbra*, the acoustic R-pattern occurs more frequently at dusk, ultimately leading to the formation of a chorus, [*sensu* [16, 43]]; the latter has been proven to be a reliable natural indicator of the species' spawning behavior [17]. At sea, *S. umbra* also produces a variable number of sounds lacking any fixed repetition rate (the irregular pattern) [16].

*Sciaena umbra* sound type can be unequivocally identified thanks to its intra-sound temporal and spectral features (i.e. number of pulses *per* sound, pulse duration, pulse period, sound dominant frequency) in datasets recorded by different teams and with different equipment, in several Mediterranean regions [23, 27, 31]. Within this relatively restricted range of variation, *S. umbra* intra-sound features can be affected by environmental, anthropic, and biological factors: in fishes, pulse rate (number of pulses per unit of time) and sound fundamental frequency can be influenced by water temperature [44–46]. Pulse rate variations have been observed in different fish species [38, 47, 48]; in *S. umbra*, this appears also to be affected by the exposure to multiple boat passages, suggesting that this species can compensate human-generated disturbance by increasing sound production [30]. Further, the competition for acoustic resources within a vocal fish community has been proven to influence the spatial and diel pattern of emissions, as well as intra-sound features [27, 49]. In fact, *S. umbra* realized acoustic niche varies across locations among the same habitat type; despite this, this species appears less efficient than others (e.g. *O. rochei*) in allocating different temporal and spectral resources across sites in relation to the presence of other vocal competitors [27].

In animals, rhythmical call properties are known to be shaped by the physical constraints (environment, individual resources) and necessities to convey information [8]; investigating temporal periodicity in animal vocalization can provide insight into the relationship between environment and the species behavior from an evolutionary perspective. As a consequence,

changes in the rhythmical properties of fish calls in relation to different environmental, biological, and anthropic conditions should be better studied. A previous study shows that *S. umbra* R-calls recorded in different locations and different years seem to be uttered with similar beats [20]; by expanding this original dataset to four different study areas and, for one location, to two different study periods, the present paper aims to describe the rhythmical patterns of *S. umbra* calls across a range of environmental conditions in the Mediterranean basin and discuss the implication of the observed results.

## Materials and methods

### Acoustic data collection

Acoustic data collection relates to previous projects [16, 17, 24, 27, 30, 50] carried out in four Mediterranean sites (Fig 1), which differed for several environmental and anthropic conditions. Data were recorded in Palma Bay Marine Reserve (Mallorca, Spain) and the Underwater Biotechnological Park of Crete (Greece). These two sites were located at 20 m depth over *Posidonia oceanica* meadows but Mallorca was characterized, overall, by a more intense anthropic presence at sea than the Underwater Biotechnological Park of Crete (Greece) (i.e. higher boat noise levels, [27]). Two other sites were considered: one in the Northern Adriatic Sea, i.e., the highly anthropized Venice inlets connecting the Venice lagoon with the sea, and the other in the Miramare Marine Protected Area, an urban MPA located in the Trieste Gulf (Italy); both sites were characterized by artificial rocky reefs close to a muddy bottom with a maximum depth of 18 and 30 meters, respectively. Data were collected in 2017 in Mallorca and Crete, in 2019 and 2020 in Venice. Two acoustic recordings were carried out in Trieste; the first in 2009 and the second ten years after (2019–2020). In each recording site, water temperature data were available (Table 1). Furthermore, the richness of the fish acoustic community (i.e. acoustic richness: number of different fish sound types) was 8 in Crete, 4 in Mallorca [27] and 2 in Venice and Trieste [17, 24, 50]. No permits were needed to conduct this research. Recordings were obtained by using the strictly observational and non-invasive technique of passive acoustic monitoring (PAM). PAM involves the deployment of acoustic data loggers in the marine environment; no animals nor animal samples were collected, and no animal procedure was carried out. Different predictions were initially made about the potential influence of these environmental conditions on *Sciaena umbra* rhythms (see S1 Table in S1 File).

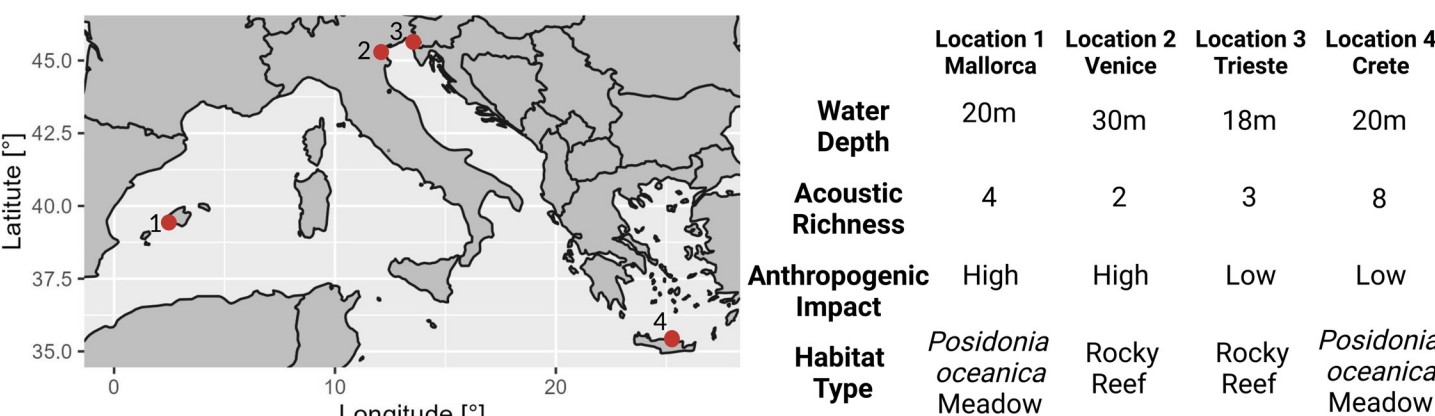

| | | Location 1 Mallorca | Location 2 Venice | Location 3 Trieste | Location 4 Crete |
|---|---|---|---|---|---|
| **Water Depth** | | 20m | 30m | 18m | 20m |
| **Acoustic Richness** | | 4 | 2 | 3 | 8 |
| **Anthropogenic Impact** | | High | High | Low | Low |
| **Habitat Type** | | *Posidonia oceanica* Meadow | Rocky Reef | Rocky Reef | *Posidonia oceanica* Meadow |

**Fig 1. Map of the study sites with information on site variability.** Map imported from the public domain Natural Earth project, using ggplot::map_data("world") in R.

**Table 1. Coordinates of recording sites; year, month and time of recordings are provided; water temperatures fall in the specific range in which *S. umbra* produces sounds (i.e. between 15 and 28˚C).**

| | Coordinates | Year | Months | Time | Water temperature (mean ± SD) | |
|---|---|---|---|---|---|---|
| Location 1 Mallorca (Spain) | 39˚ 27.885' N | 2017 | July-August | 9 p.m.– 10 p.m. | 24–27˚C | |
| | 02˚ 43.331' E | | | | (26.7±0.9) | |
| Location 2 Venice (Italy) | 45˚ 19.899' N | 2019–2020 | June-August | 8 p.m.– 11 p.m. | 21–28˚C | |
| | 12˚ 20.114' E | | | | (24.1±3.2) | |
| Location 3 Trieste (Italy) | 45˚ 42.133' N 13˚ 42.700' E | 2021 | July-August | 7 p.m.– 0 a.m. | 21–26˚C (25±1.2) | |
| Trieste (Italy) | 45˚ 42.133' N 13˚ 42.700' E | 2009 | July-August | 9 p.m. - 3 a.m. | 21–25˚C (22±1.2) | |
| Location 4 Crete (Greece) | 35˚ 20.749' N 25˚ 16.722' E | 2017 | July | 8 p.m.– 10 p.m. | 25–26˚C (25.9±0.6) | |

SNAPs dataloggers (Loggerhead Instruments, FL, USA), connected to an HTI-96-min hydrophone (sensitivity: 170 dB re 1V μPa-1; frequency range 2 Hz– 30 kHz; recording .wav files at 44.1 kHz, and 16 bits) were used for collecting acoustic data at Palma Bay Marine Reserve (Mallorca, Spain) and at the Underwater Biotechnological Park of Crete (Greece); here, the SNAPS were synchronized before deployments and were set for recording 1 min every 11 min (over the 24 h) at 44.1 kHz and 16 bits for one month [27]. In Venice (Italy), a pre-amplified GP1280 hydrophone (Colmar SRL, La Spezia, IT; sensitivity −170 dB re. 1 V Pa$^{-1}$; frequency range, 5 Hz–90 kHz) connected to a Tascam Handy Recorder (Tascam, Montebello, CA) generating .wav files at 44.1 kHz and 16-bit depth was used; three single 10-h acoustic summer campaigns were conducted at 40 listening points along two summers [17, 24, 50]. In Trieste, recordings were collected (i) by using an acoustic data logger prototype provided with a pre-amplified TC 4013 hydrophone (Reson, A/S, Salangerup, Denmark; sensitivity −170 dB re 1 V μPa$^{-1}$; frequency range 1 Hz to 170 kHz) connected to a Gemini iKey Plus Recording Device (10 min .wav files at 44.1 kHz and 16 bits) in 2009 (series of nocturnal 11-h recordings from June-September) and (ii) by using an underwater acoustic data logger SNAP settled for recording 1 min every 5 min (over the 24 h) for one week (unpublished data).

## Acoustic and rhythm analysis

The acoustic samples were analyzed by audio and visual assessment using Raven Pro 64 1.4 (Bioacoustic Research Program, Cornell Laboratory of Ornithology, Ithaca, NY, USA; sound files down-sampled to 4 kHz, fast Fourier transform (FFT) size 256 points, 50% overlap, Hanning window).

In total, five datasets were considered (Table 1). Twenty 1-minute sequences of fish sounds were extracted *per* dataset. Fish sound sequences are defined as the temporally ordered repetition of at least three elements (one pulsatile sound surrounded by silence), which are temporally clearly isolated from preceding and following elements [20]. A sequence was confidently assigned to the same individual if all elements within the sequence were characterized by the same Signal-to-Noise Ratio (assuming the fish is stationary).To reduce the risk of pseudo-replications (i.e. sampling the same individual more than once), each 1-minute sequence was obtained by a non-consecutive recorded audio file in each dataset; therefore, the assumption was that each sequence was emitted by a different individual.

Ten additional 10-minute sequences of sounds have been extracted from the Trieste 2009 dataset. These sequences were analyzed in accord with the case of 1-minute sequences and the same individual calls were tracked along the file based on a similar Signal-to-Noise Ratio of the identified R-calls but, given the longer duration of the recordings, a sequence was considered "terminated" after 10 seconds of silence. In its turn this implies that, in some of these 10-min

recordings, multiple sequences could be assigned to the same individual (Fig 2), thus allowing to characterize intra-individual rhythm variability. Unfortunately,10-minute recordings were available only for Trieste 2009; as a result, the here considered sample size is very small and presents strong temporal dependencies within one recording; therefore this analysis should be considered preliminary.

Per each identified sequence, any present sound (i.e., element, see [20] for details about terminology and procedure) was selected in the oscillogram from its onset to its offset using the selection function in Raven, which generates a selection table. In the Raven selection table, each selection made by the operator (i.e. each element) is labelled by its start and end times and lower and upper frequency. The Raven selection tables were then exported for subsequent analyses.

For rhythm analysis, the workflow developed by [15] was followed, as described by [20], to which we refer the reader for specific details. The interval between consecutive sounds (inter-onset-interval; IOI) of a sequence was calculated and the distribution of IOIs was visually represented using histograms. The coefficient of variation (CV; the ratio between standard deviation and sample mean) and the best-fitting frequency were calculated for each sequence. In particular, the best-fitting frequency describes a perfect underlying isochronous pattern, that best matches the element onsets in a sequence. It is reported as a frequency (Hz), in elements per second. An IOI beat of 5 Hz would indicate an underlying perfect isochronous pattern with 5 elements per second. This means an element every 200 ms best matches the onset of elements and can therefore be used as an adequate description of the temporal structure of the sequence. The custom-made app to run the rhythm analysis can be found at: https://github.com/LSBurchardt/R_app_rhythm/tree/master/RhythmAnalysis).

## Environmental variables and statistical analysis

Different environmental, biotic and anthropogenic variables were taken into consideration due to their proven potential influence on *S. umbra* vocal behavior (see Introduction), i.e., water temperature, recording month, time of day, acoustic richness, and vessel density for

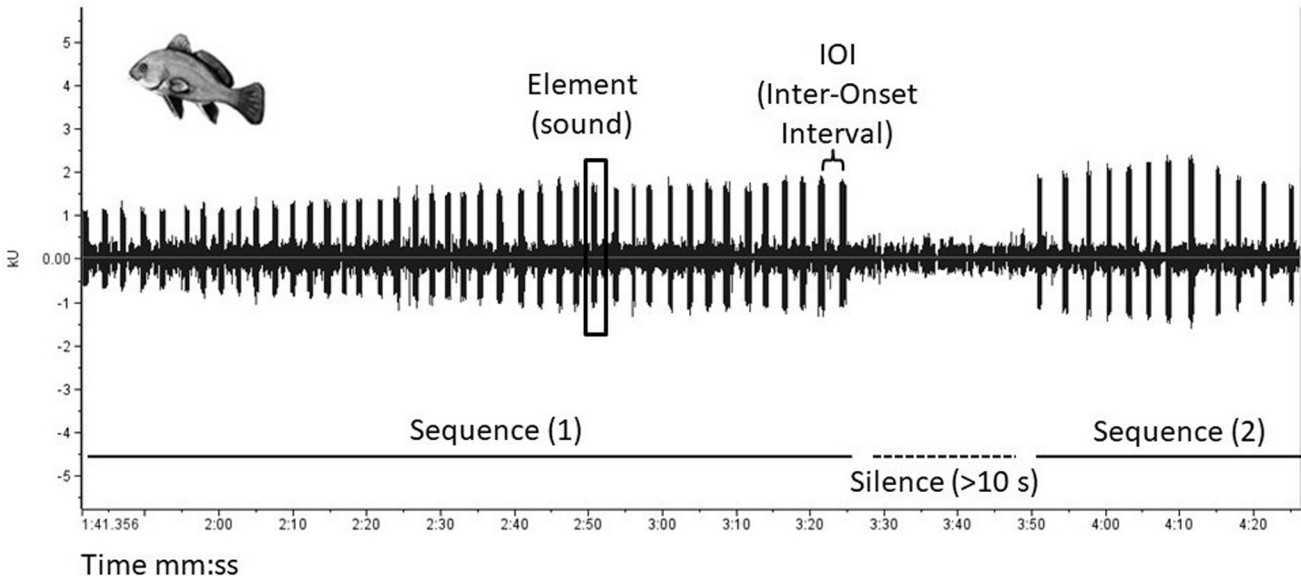

**Fig 2. Exemplary oscillogram.** Two sequences from a 10-minute file can safely be assigned to the same individual due to a very similar Signal-to-Noise Ratio.

different vessel types. To estimate vessel traffic in the study areas we used the Automatic Identification System (AIS) available data; generally speaking, these data include position and identification of ships of 300 gross tonnage and upwards, cargo ships of 500 gross tonnage and upwards and all passenger ships; smaller recreational boats typically do not use AIS. A square of 25 km$^2$ around each monitoring location (center) was considered; the average AIS traffic density was extracted from the EMODnet Human Activities web portal (www.emodnet-humanactivities.eu). EMODnet maps are based on AIS data yearly purchased from Collected Localization Satellites (CLS) and ORBCOMM; here, density is expressed as hours *per* square kilometer per month in 1x1 km cells and it is provided for different ship types. Data were available by month of the year and were downloaded for the months corresponding to the study's acoustic recordings. Six different vessel categories were considered separately: trade vessels, fishing vessels, recreational vessels, passenger vessels, "other" vessel types and all vessels combined.

Linear relationships were not verified for all variables, for example, because of many similar values and zero inflation, and therefore a linear approach was not suitable. General Additive Mixed Models were therefore used to quantify the influence of environmental, biotic and anthropogenic factors on the IOI beat. GAMs allowed both categorical and continuous variables to be included in the model. Continuous variables were smoothed. For details on the modelling approach see S2 Table in S1 File.

One outlier (3x standard deviations away from the mean for the IOI beat [Hz] parameter) was found and removed from the data before running the model. Model selection was based on the Akaike Information Criterion (AIC; [51]). In a step-wise process, parameters with the worst significance were removed from the model, until the best AIC was reached (S2 Table in S1 File). The statistical significance of the remaining factors was determined by a *P*-value of ≤0.05. The analysis was performed in R 4.2.3 (R Core Team, 2023).

## Results

An isochronous rhythm was found in all locations. This is confirmed by a steep unimodal distribution of Inter-Onset-Intervals (Fig 3A), as well as by the low coefficient of variation within sequences (Fig 3B) for all locations and different years (mean CV: 0.18, min = 0.08, max = 0.53, SD = 0.09).

The best-fitting IOI beats were similar for most study areas, ranging from 0.016 and 0.52 Hz with an average of 0.36 Hz and a standard deviation of 0.08 Hz (Fig 3C). The data collected at the Venice inlets, however, represent an exception since the IOI beat was significantly faster (Fig 3C, 0.46 Hz, min = 0.32 Hz, max = 0.68 Hz, SD = 0.1 Hz) than all the other datasets, (ANOVA: F = 6.647, p = 0.0005***; Student's t-Test with Bonferroni correction for multiple testing and Effect Size Cohens D: Trieste vs. Venice p = 0.0014, D = 1.18; Crete vs. Venice p = 0.0014, D = 1.23; Mallorca vs. Venice p = 0.004, D = 0.99).

Approximately 67% of the observed variation in the IOI beat could be statistically explained by eight variables according to GAMs (sample size N = 77; Adj. $R^2$ = 0.67, Deviance explained = 77.2%, GCV = 0.004, Scale est. = 0.003, Table 2 and S2 Table in S1 File, S1 Fig in S1 File). These eight variables are day of the recording, the interaction between day and month of the recording (and therefore the breeding season progression), the time of day [h], total vessel density, vessel density of trade ships, vessel density of fishing ships, acoustic diversity and temperature.

Of the eight variables that contribute to explaining 67% of the variance in the sample, only four variables had a significant effect on the IOI beat [Hz]. These variables were the interaction between the month and day of the recording, trade and fishing vessel density and acoustic

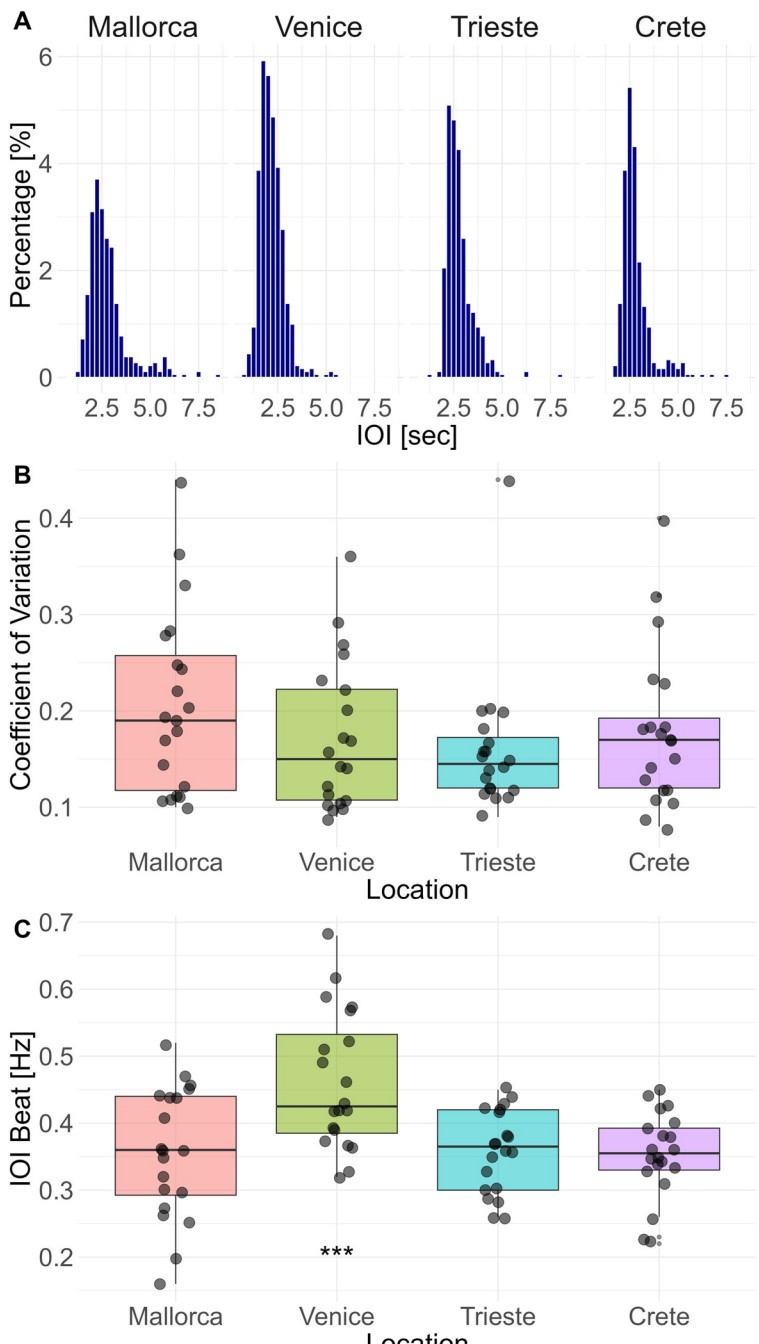

**Fig 3. Overview of rhythmic parameters.** (a) Histogram of the duration of Inter-Onset-Intervals (IOIs) for the *S. umbra* sequences analyzed per location. (b) The coefficient of variation for the *S. umbra* sequences was analyzed per location. (c) IOI Beats [Hz] for the *S. umbra* sequences analyzed per location.

richness. The interaction between the month and day of the recording had a statistically significant influence at the .001 level on the IOI beat as determined by the GAMs model (S2 Table in S1 File). It seems, therefore, that IOI beat increases as the reproductive season proceeds(higher in August vs. July and June; Fig 4). Further, the *S. umbra* IOI beat seems to be influenced (i.e., speed up) by the number of sound types present in each location (i.e. acoustic richness,

**Table 2. Summary of the General Additive Mixed Model (GAMs) results.** Model variables after model selection, for smoothed values, the F statistic and p-value are reported. For categorical values, the Estimate and p-value are reported. Significance levels: * p ≤ 0.05; ** p ≤ 0.01; *** p ≤ 0.001.

| Coefficient | Estimate | F | p |
|---|---|---|---|
| (Intercept) | -0.36 | -- | 0.16 |
| Day | -- | 2.33 | 0.053 |
| Interaction Day by Month | -- | 5.12 | 0.001** |
| Time [h] | -- | 2.08 | 0.16 |
| Acoustic Diversity | 0.02 | -- | 0.00*** |
| Vessel Density Total | -- | 3.09 | 0.09 |
| Vessel Density Trade | -- | 28.91 | 0.00*** |
| Vessel Density Fishing | -- | 14.32 | 0.00*** |
| Temperature | -- | 2.03 | 0.06 |

p = 0.00***). Finally, the model shows statistical significance for the vessel density of trading ships (p < 0.00***) and fishing ships (p < 0.00 ***). Conversely, the total vessel density does not have a significant effect, suggesting that specific vessel types might influence rhythm production more than others. Both, higher trading and fishing vessel density speed up the IOI beat (S2 Fig in S1 File and Fig 3). Vessel density distribution *per* location is shown in Fig 5. The temperature did not have a clear effect on the IOI beat (Fig 6).

In contrast with the inter-site variability, within-area variability over long periods was not found: in fact, similar IOI beats were found in Trieste in 2009 and 2021 (Fig 7, Welch two sample t-test: t = -0.45, df = 34.99, p = 0.65, effect size Cohens D: 0.15).

Intra-individual variability in IOI beat was further considered on the basis of ten 10-minute files collected in Trieste in 2009; out of them, only three files had multiple sequences assigned to single individuals (see Materials and Methods for details): in one file, four vocal sequences were assigned to one individual, in two other files two vocal sequences were assigned *per* individual. Beat frequency range size, i.e. the difference between the fastest and slowest beat (Hz) calculated per *each* of the three individuals, was equal to 0.17, 0.006 and 0.08 Hz respectively. For comparisons, the beats range size in the whole 1-minute dataset (n = 80) was 0.33, at least two-fold bigger than the one measured within individual calls. As a consequence, IOI beat variability seems to be not driven by intra-individual but mostly by inter-individual differences. This conclusion however should be considered with care given the small sample size used for the analysis.

## Discussion

Vocal activity is an important communication modality in fish: it reflects vital processes [41, 52–59] and conveys species-specific, individual characteristics and motivation states [60–63] under the pressure of the environmental constraints, such as interference by ambient noise patterns and sound transmission properties [64–66]. Here, *S. umbra* reproductive R-calls were recorded in four sites across the Mediterranean basin (from West to East: Mallorca, Venice, Trieste, Crete) and their rhythm was analyzed. Given the highly variable environmental conditions, ranging from artificial reefs on muddy bottoms to *Posidonia oceanica* meadows, as well as in local soundscape (due to different fish community composition and different human-generated noise inputs), differences were expected between locations. On the contrary, a highly stereotyped isochronous rhythm of around 0.36 Hz (0.36 elements per second) was found in all sites except for Venice, where the R-calls presented a faster beat. Additionally, an isochronous rhythm, with a consistent beat, was found in the *S. umbra* population recorded at

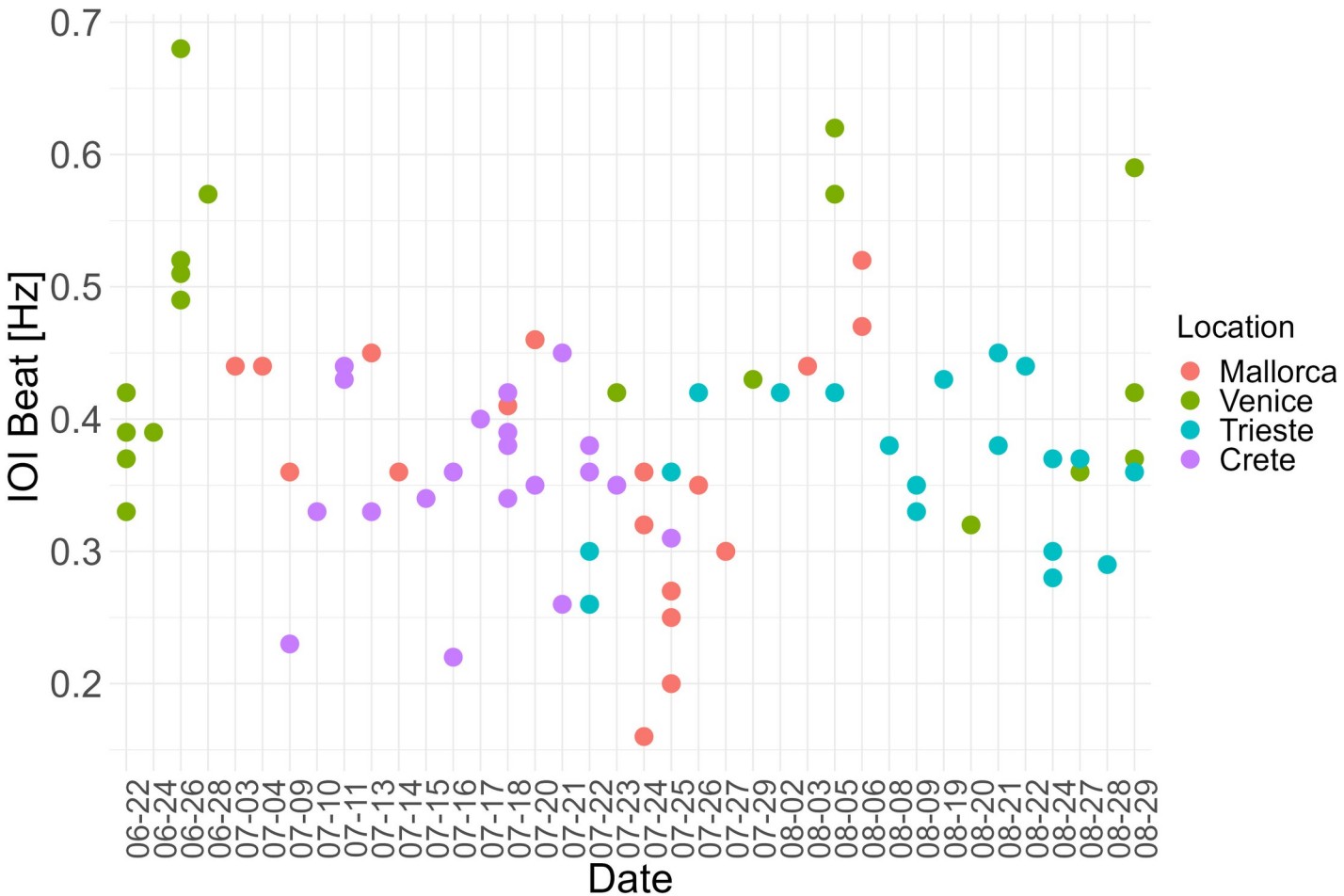

**Fig 4. Inter-Onset-Intervals (IOI) beat distribution across recording dates.**

the Marine Protected Area of Miramare (Trieste, Italy) about 10 years apart. In particular, the preliminary analysis on these 10-min R-sequences indicates that the beat variability seems to be not driven by within-individual but most likely by inter-individual differences, suggesting a consistency of this parameter at the individual level. If this is the case, a relatively stable population of individuals are expected to show a similar pattern of beat calls in time, as was the case of the Miramare Protected Area given the nature of protection of this area and the longevity of the species, whose individuals can live up to about 15–20 years [67].

All together the collected evidence indicates that the stereotypical R-calls produced by *S. umbra* males during the mating season tend to maintain a species-specific, consistent rhythm across space and time. In the following sections, different potential reasons for the unexpected consistency will be discussed as well as possible explanations connected to the observed variability.

### Consistency of *S. umbra* R-calls isochrony property between and within locations

The rhythmic behavior may have evolved in many animal species in response to pressures for mate attraction as a powerful overall signal to attract females from further away [68]. In line with this theory, the observed consistency of isochronous rhythm in *S. umbra* calls recorded in

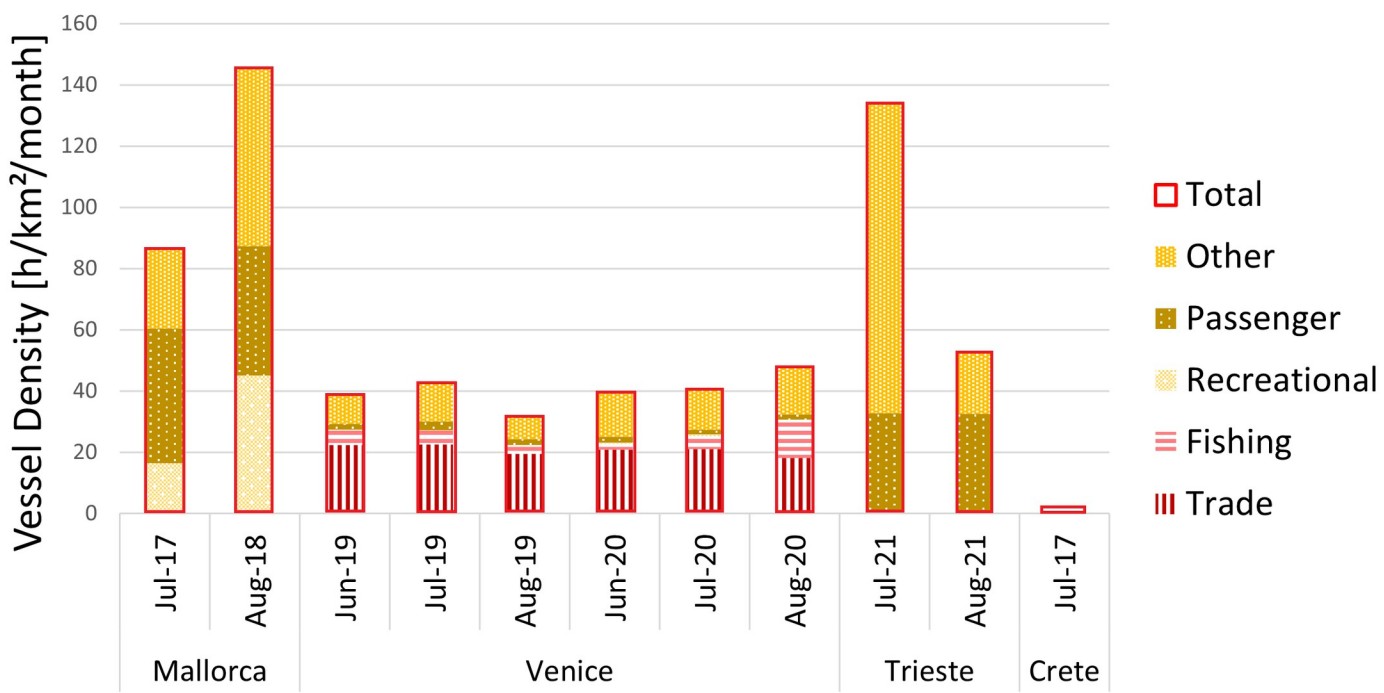

**Fig 5. Vessel density per location and recording month.** Vessel types that were part of the final GAM model are indicated in red colors and stripes; vessel types cut from the model during model selection are indicated in yellow colors and dots, the total vessel density, which is also included in the final model is indicated as the red boundary. Locations are sorted from West to East, indicated are the month and the year of the data collection (e.g., Jul-17 indicates the vessel density for July 2017).

different habitats leads to hypothesize that this temporal property of the calls is a fitness-related acoustic trait, i.e. an important trait for the individual and therefore the species' reproductive success. Our hypothesis is supported by many considerations.

First, scientific evidence suggests that fish generate, perceive, and recognize acoustic regularity in time. Some fish sounds consist of pulsed sounds with a distinct temporal patterning [61]; intra-sound timing (number of pulses, pulse repetition rate) has been proven to encode relevant information [62, 69–72]. On the other hand, nothing/less is known about the role of inter-sound rhythmical patterns, which are the focus of this present study. Generally speaking, fish's auditory system is well suited for temporal processing [73]; teleost temporal resolution abilities are defined in the order of a few milliseconds or even less, mostly in accordance with mammals and birds. This implies that some species at least can discriminate each pulse within a sound [74, 75]. Further, sound temporal patterns, as well as amplitude fluctuations and frequency content, are reflected at the level of the brainstem, so that fish's auditory system can accurately encode the temporal patterns of conspecific sounds [76–78].

Temporal coding of relevant information is mostly expected in fish whose sounds consist of a series of pulses with little modulation in frequency; this is the case Sciaenids, with few exceptions such as the ones reported by [48, 79, 80]. In this family, pulse duration, pulse repetition rate and number of pulses *per* sound vary among species, leading to conclude that discrimination between species or individuals is based on call temporal characteristics [40, 41, 80–84]. The relevance of temporal features in Sciaenids is also due to the physiological characteristics of its sound-producing mechanism, which, in *S. umbra*, is present in males only: the sound-generating mechanism involves a pair of super-fast, extrinsic sonic muscles acting on the swim bladder [31]. In Sciaenid's sounds, a single pulse corresponds to a unit of muscle contraction,

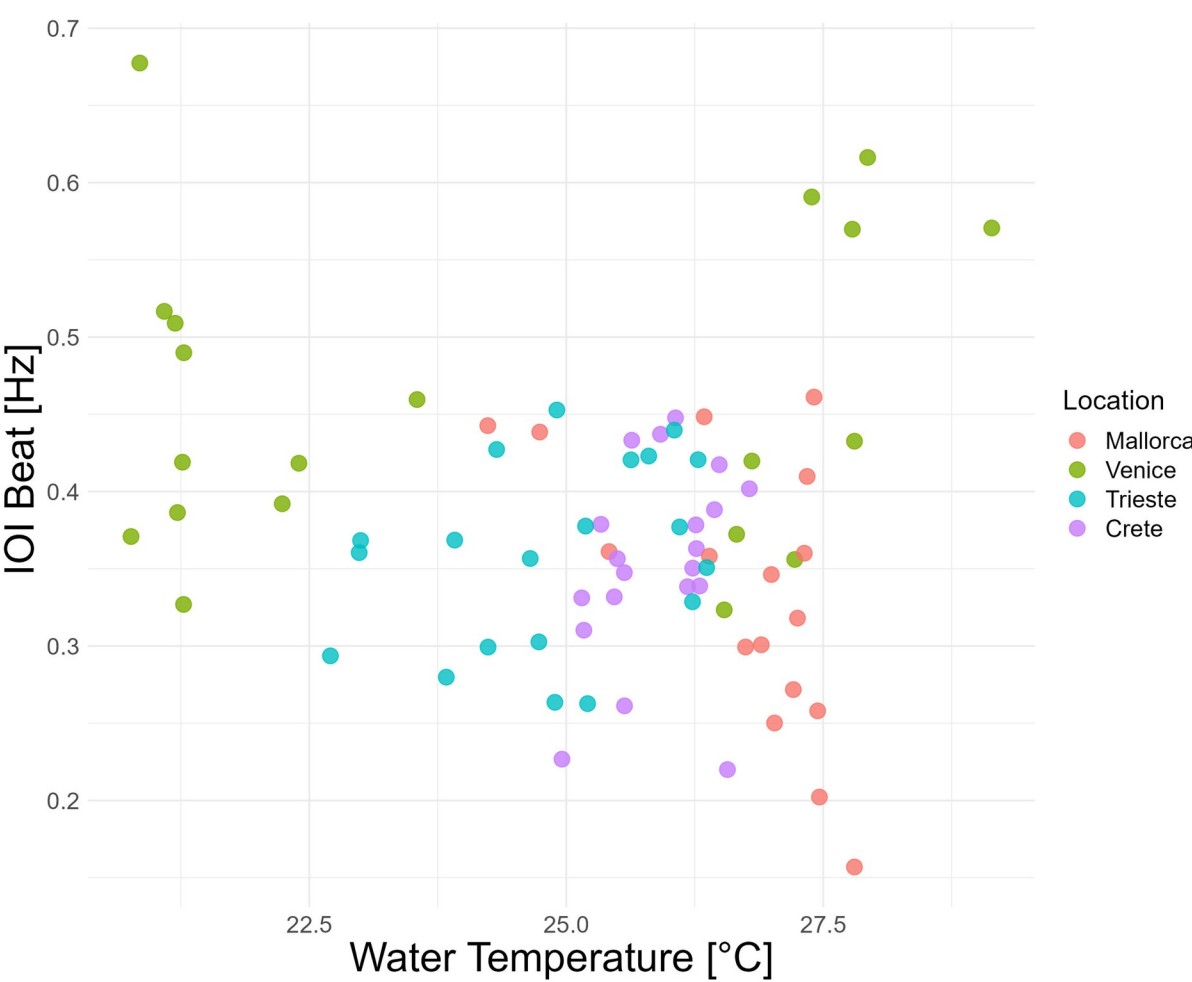

**Fig 6. Inter-Onset-Intervals (IOI) beats [Hz] *vs.* water temperature; recording locations are highlighted by different colors.**

with the cycle of each pulse giving the fundamental sound frequency (i.e. base frequency of which harmonic components exist) [85]. When considering peak frequency (i.e., the frequency with the highest energy), this shows a relatively low variability in relation to fish size in comparison to other fish taxa [86]. Overall, intra-sound feature consistency has been found in both space and time [31].

Finally, temporal features are known to be relevant during Sciaenid reproduction; the formation of seasonal reproductive aggregations at sea and call rate appear to be closely related [40]. Furthermore, the time in which the number of calls peaks (i.e., time of day) was found to be positively related to the timing and number of eggs collected at sea [47, 87, 88]. Captivity studies proved that call temporal features reflect spawning activity in many species: in *Scienops ocellatus* an increase in the number of calls, with longer calls and more pulses, has been recorded when spawning occurs [18, 89] and *Cynoscion nebulosus* is more likely to spawn when male fish call rate is higher [90]. In captivity, during spawning nights *Argyrosomus regius*, *Sciaenops ocellatus* and *Umbrina cirrosa* emit more calls per unit of time(*versus* prior to spawning); furthermore, calls are significantly longer and characterized by a higher number of faster-repeated pulses [18]. These same temporal features (i.e. longer sounds, made of a higher number of faster-repeated pulses) were found during the *S. umbra* chorus recorded at sea [17].

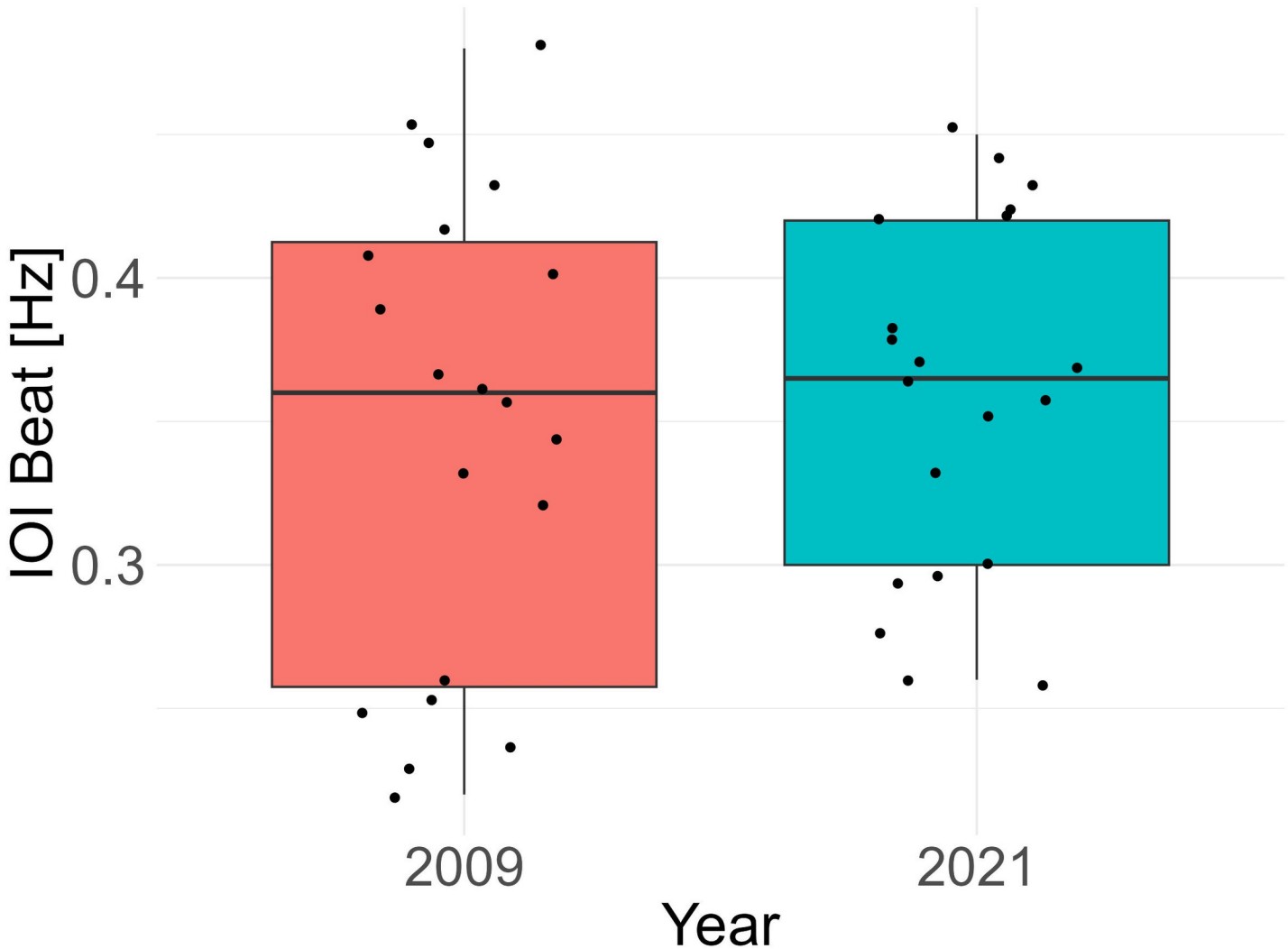

**Fig 7. Inter-Onset-Intervals (IOI) beats [Hz] for the *S. umbra* sequences analyzed in the same location (Trieste) about 11 years apart: Rhythm frequency was consistent over time, and no significant difference was found.**

Given all this evidence, inter-sexual selection is expected to action *S. umbra* rhythm in the R-call pattern, supporting our original hypothesis.

## Variability of beat frequency in the isochronous R-pattern between and within locations

Spatial variability was found in the beat frequency: *S. umbra* R-calls collected along the Venice inlets were characterized by a significantly higher beat frequency. It is interesting to notice that other parameters, such as the coefficient of variation between intervals are comparable to the other locations. The reason for a higher beat frequency is therefore not to be found as a bias of higher variability within sound sequences.

The Venice inlets allow navigation from the Venice lagoon to the Adriatic Sea and *vice versa*. Intrinsically, they are characterized by high traffic of cargo, passengers and fishing vessels, as well as conspicuous distribution of recreational and small-sized boats [91]. In turn, this substantially increases the inlets' background noise, resulting in levels typical of harbor

underwater areas [24, 92].We therefore hypothesized that this effect could be connected with the highly anthropized local soundscape.

A General Additive Mixed Model was considered to statistically analyze the influence of the potentially relevant environmental and anthropogenic factors on *S. umbra* beat frequencies. About 67 percent of the observed variability in the R-sound beat could be explained by considering four variables: the interaction between the day and month of the recording, the presence of other fish sound types, i.e., acoustic competition in the study area, and vessel density (here used as a proxy of fish exposure to human-generated noise) of trade ships as well as fishing boats. These results indicate a change in IOI in Venice recordings mostly due to the high vessel density of fishing and trade and different recording times compared to the other locations.

*Sciaena umbra* vocal activity is known to be characterized by seasonal and daily variations [16, 23, 31] so the influence of the recording timing on the R-pattern was expected, in contrast with intra-sound features [44] the water temperature seems to not play any role in the beat property of this calling. On the other hand, it is known that the noise generated by maritime traffic negatively affects marine fauna at various levels, including animals' behavior and acoustics (reviewed by [93–95]). More specifically, multiple boat passages have been found to elicit an increase in the *S. umbra* pulse rate (pulses *per* minute) recorded at sea [30]. The faster rhythm of *S. umbra* sound production observed in recordings characterized by a dominant presence of vessel noise (i.e. in the Venice inlets) is consistent with the previous research since speeding up the R-pattern rhythm will necessarily result in a higher number of pulses produced in a given time.

Finally, the presence of inter-specific acoustic competition is suggested to play a potential role in influencing *S. umbra* R-rhythm beats. [27] proved a reduced inter-site variability when considering intra-sound temporal features, which could result from constraints associated with the species' sound-producing mechanism [31]. The hypothesis is that species with a relatively reduced degree of sonic system plasticity, and consequently, with a relatively reduced degree of variability in intra-sound features resources allocation, resort to rhythmical variability to compensate for increased vocal competition. A faster R-rhythm would potentially compensate for inter-specific masking generating a signal redundancy, with a mechanism that is potentially similar to human-generated noise masking: in fact, many forms of vocal adjustment are gendered in animals in case of impaired communication including the production of redundant signals [96]. According to communication theory [97], vocal compensation is predicted when the balance between the benefits of successful communication and the costs of such a vocal change is positive, as is the case here, given the key role of R-pattern rhythm in *S. umbra* reproduction discussed in the previous paragraph.

Despite all these considerations, it has to be kept in mind that the present analysis was based on opportunistic acoustic data collected in the context of other projects (see Materials and Methods for details) and therefore, the presented results reflect the available datasets; on the other hand, not all variability observed in the beat data could be statistically explained by using the considered variables. This also means that more data are needed and/ or that other factors can potentially affect *S. umbra* R-rhythm as, for example, local population structure (i.e. the local fish density, male/female ratio, age of callers, and so on) and/or the single fish behavior during calling activity. Unfortunately, these features could not be quantified in the present study. In turn, this calls for further studies. In particular, keeping the influence of recording time within the season in mind, it would be very interesting to record data within one location throughout the season, to get a detailed view of rhythmic properties across the breeding cycle in *S. umbra*.

## Conclusions

In conclusion, *S. umbra* R-calls showed an isochronous nature that was independent of environmental features indicating it is a fitness-related trait for the species. This is in accordance with the case of the Lusitanian toadfish (*Halobatrachus didactylus*); here, only males with high-quality conditions could sustain isochronous calling [19, 98] and, therefore, isochrony was suggested to act as an honest signal of male quality. In *S. umbra*, a degree of plasticity was found when considering the beat frequency, which varied spatially, likely influenced by the temporal-related dynamics along the species' reproductive season, as well as by competition with other biological and human-generated acoustic sources. Overall, this paper shows that the assessment and comparison of vocal rhythm represents a critical aspect of fish communication, and calls for more investigations on this topic.

## Supporting information

**S1 File. Supplementary information containing details on theoretical considerations and modeling.**
(DOCX)

## Acknowledgments

We would like to thank; i) Prof. Eric Parmentier for providing resources and equipment which allowed the collection of three datasets presented in this study; ii) Prof Stefano Malavasi, Dr. Riccardo Fiorin, Dr. Federico Riccato and Dr. Chiara Facca, Dr. Saul Ciriaco and Dr. Maurizio Spoto, Dr. Antonio Codarin for allowing the data acquisition with resources and field-work support in Venice and Trieste (Italy), respectively; iii) Dr. Ignacio A. Catalan and Dr. Josep Alós for field support in Mallorca (Spain); iv) Dr. Thanos Dailianis and Dimitris Androulakis for field support in Crete (Greece); v) Dr. Antonio Petrizzo for his help in extracting vessel traffic data; and vi) Dr. Kathryn Turnbull for insightful discussions on statistical treatment.

## Author Contributions

**Conceptualization:** Marta Picciulin, Marta Bolgan, Lara S. Burchardt.

**Data curation:** Marta Picciulin, Marta Bolgan.

**Formal analysis:** Marta Picciulin, Marta Bolgan, Lara S. Burchardt.

**Investigation:** Marta Picciulin, Marta Bolgan, Lara S. Burchardt.

**Methodology:** Marta Picciulin, Marta Bolgan, Lara S. Burchardt.

**Project administration:** Marta Picciulin, Marta Bolgan, Lara S. Burchardt.

**Software:** Lara S. Burchardt.

**Supervision:** Marta Picciulin, Marta Bolgan, Lara S. Burchardt.

**Validation:** Marta Picciulin, Marta Bolgan, Lara S. Burchardt.

**Visualization:** Lara S. Burchardt.

**Writing – original draft:** Marta Picciulin.

**Writing – review & editing:** Marta Picciulin, Marta Bolgan, Lara S. Burchardt.

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
