## [Decision Letter · Decision Letter 0]

6 Sep 2023

PONE-D-23-21637Rhythmic properties of Sciaena umbra calls across space and time in the Mediterranean SeaPLOS ONE

Dear Dr. Burchardt,

Thank you for submitting your manuscript to PLOS ONE. After careful consideration, we feel that it has merit but does not fully meet PLOS ONE’s publication criteria as it currently stands. Therefore, we invite you to submit a revised version of the manuscript that addresses the points raised during the review process.

Please see the comments from two reviewers below. Both reviewers seem positive about the contributions of the work, and have provided suggestions for clarifications or ways to strengthen the manuscript. We now invite you to respond to these either by incorporating revisions or rebutting specific concerns.

We look forward to receiving your revised manuscript.

Kind regards,

Hanna Landenmark

Staff Editor

PLOS ONE

Journal Requirements:

4. We note that [Figure 1] in your submission contain [map/satellite] images which may be copyrighted. All PLOS content is published under the Creative Commons Attribution License (CC BY 4.0), which means that the manuscript, images, and Supporting Information files will be freely available online, and any third party is permitted to access, download, copy, distribute, and use these materials in any way, even commercially, with proper attribution. For these reasons, we cannot publish previously copyrighted maps or satellite images created using proprietary data, such as Google software (Google Maps, Street View, and Earth). For more information, see our copyright guidelines: http://journals.plos.org/plosone/s/licenses-and-copyright.

Reviewers' comments:

Reviewer's Responses to Questions

**Comments to the Author**

1. Is the manuscript technically sound, and do the data support the conclusions?

Reviewer #1: Partly

Reviewer #2: Yes

2. Has the statistical analysis been performed appropriately and rigorously? 

Reviewer #1: I Don't Know

Reviewer #2: Yes

3. Have the authors made all data underlying the findings in their manuscript fully available?

Reviewer #1: No

Reviewer #2: Yes

4. Is the manuscript presented in an intelligible fashion and written in standard English?

Reviewer #1: Yes

Reviewer #2: Yes

5. Review Comments to the Author

Reviewer #1: The authors present interesting and through study of Sciaena umbra acoustic signaling behaviour across four different sites in the Mediterranean. The paper is generally well written and follows a clear, linear structure and narrative. I enjoyed reading the manuscript and would like to thank the authors and the editor for giving me the opportunity to comment on this work.

I think that overall the methodology is solid and the results are well presented. The topic of acoustic communication in fish is definitely understudied, especially when taking into the consideration the richness of fish species and their enormous economic value. I also appreciate that the authors choose to explore the rhythmic structure of the signals, this is indeed a very interesting avenue and a potential fitness indicator which started receiving more attention only recently. So, the paper is timely and offers few great suggestions for future work. I do have few comments but think that the authors should be able to address them with relative ease.

Major comments:

1. One major issue that immediately popped out when first reading the paper is the sub setting of the collected data (L159). The authors selected “Twenty 1-minute sequences …. per dataset”. Since the data was collected by passive acoustic monitoring, over several months, I assume that much more acoustic data was collected. I am very well aware that processing and annotating audio data is extremely time and labor consuming, however I think that the authors should give a more detailed justification for their chosen sample size. Especially I would be very interested in seeing an estimation of the overall call time per site, as it can indicate population density and perhaps have an effect on the signaling behavior.

2. Another related concern would be the uniformity of sampling throughout the data collection periods. Since the authors mention that breeding season (August) is associated with an increase in signaling rhythm, they might want to provide some information on how balanced their sampling was. Looking at Fig4 it seems that most of the August data is coming from the Trieste site. Venice site has half of its data in June, large gap in July and more data in August again. Just from eyeballing the scatter plot, Trieste shows an increase in rhythm but other sites might not. Adding more July samples from Venice might strengthen the regression line for the month effect. I wonder if the authors considered fitting a model with interaction terms and random effects to allow a better fit and a better explanation of the relationships among regressors. Or they could try and balance the dataset by making sure that the 4 sites are more or less equally represented and sampled periodically throughout the period of interest.

3. Vessel traffic analysis: In L217 the authors mention that 6 different vessel categories were considered but the model output (Table2) only shows trade and fishing vessels. Were the other vessel categories omitted in the model selection process? From Fig5 it seems that the trade and fishing categories were mainly represented on Venice site. This site also had the lowest acoustic richness and highest water depth. So, as authors clearly mention in the discussion (L438) the effect if the vessel noise is very inconclusive and other factor could be driving the higher signalling rhythms in their study species. Reading the Abstract I expected a much stronger support for the anthropogenic effects on the signaling rhythm but after going through the results, I no longer see this as as a major finding of the study. Perhaps the authors would consider toning it further down in the Abstract.

I have a perhaps naïve suggestions (since I know very little about marine vessel noise pollution levels), would it make sense to use the tonnage of the vessel as a proxy for noise levels instead of the vessel type? It might provide one continuous variable, instead of 6 categories, potentially improving statistical power. This is off course only relevant if larger vessels actually generate more noise pollution.

4. L413 here the authors mention that 27% of the rhythm variability are explained by the three factors (month, presence of other fish and vessel density). In L258 the same 27% are attributed to the month of the recording (the breeding season) only. Overall it was not clear to me what procedure the authors used to estimate the contribution of the different variables to the changes in the signalling rhythm. Also, more generally, while I am not a statistician, I think that the paper can benefit from more details on the process of model diagnostics and selection. The authors mention in L222 that the model assumptions were checked by visual inspection. I think that adding a supplement with the data, full model design, diagnostic plots and model selection steps can help the readers to better assess the data and the results.

Minor comments:

5. L79-80 This sentence might benefit from rephrasing. It is not clear variable in number of what - Rhythmic categories, pulses or variable number of sounds?

6. L102 – here authors state that “nothing is known about how the rhythmic properties of fish sounds change…..” but the previous paragraph actually has a very nice summary of previous work linking rhythmic properties of fish sounds to environmental and biological factors. I would tone down the claim a bit.

7. L125 – The Trieste data collection: this is not a continuous 10-year monitoring but two sampling efforts in 2009 and 2019-2020. I feel that "a time span of 10 years" is a misleading phrasing in this case

8. L128 - Is number of different sound types is indication of different fish species?

9. L199 - the word spacing is off here

Reviewer #2: Rhythmic properties of Sciaena umbra calls across space and time in the Mediterranean Sea, by Marta Picciulin, Marta Bolgan and Lara Burchardt,

It is a very interesting manuscript that shows results on the sound production rhythms of an species of Sciaenid.

These results, in my opinion, are original to be published in Plos One, although the authors have a previous publication where they partially describe these results, in this manuscript the authors detail the results in this species in greater detail. It is pioneering work that talks about the rhythms of sound production in a fish.

Although the manuscript presents a very good representation of the data and its statistical analysis, I think it is missing some citations that are important and will give the manuscript a better strength.

My detailed comments are attached.

In short, I think it is a good manuscript to be published, which needs a minor revision.

Good Luck, Javier S. Tellechea.

Introduction

Line 44-46. This sentence is not clear, 30 times compared to what?

Line 73. Like other Sciaenids (Tavolga, 1964; (Mok and Gilmore, 1983; Fine et al., 2004; Ramcharitar et al., 2006; Tellechea et al., 2010)

Line 62. Define the R pattern.

Line 76. “The acoustic R-pattern occurs more frequently at dusk…” in which species?

Not in all species the choirs are at dusk, see Tellechea et al 2011.

Line 88. See the paper about sound and wáter temperature :

Connaughton, el al 2000, Holt 2002.

Line 90-93. Here it is necessary to cite articles where periods of fish sound production are described. (Mok & Gilmore 1983; Connaughton

& Taylor 1995; Locascio & Mann, 2008

Methods

Line 144. “Colmar SRL, La Spezia, IT”, It is not clear to me why this location followed by the description of the power...

Results.

Table 2. The reason for the astrisk in the value 0.018 must be explained in the legend of the table.

Discusion

Line 310. Add quotes that are importante in fish vocal activity:

(Fine et al 1977; Lobel and Mann, 1995; Lobel 2002; Rountree et al., 2006; Luczkovich et al., 2008;

Tellechea et al., 2010; Amorim et al., 2015

Line 365. More quotes must be add: Saucier, and Baltz.(1993; .Ladich and Fine, 2006; Mok et al., 2009; Tellechea

et al., 2010a, Tellechea and Norbis, 2012

Line 368-372. yes, but there are variations such as in the case of Pogonias cromis (Locacio and Mann 2009) and Pogonias courvina (Tellechea et al 2011 and Tellechea et al 2022, where the reproduction sound is modulated... it would be interesting if they add this exception, since they are the only two species that present this characteristic.

Line 378. In addition to (Luczkovich et al. 1999), add Connaughton, M. A., and Taylor, M. H. (1995).and Tellechea, J. S., Bouvier, D., and Norbis, W. (2011)..

6. PLOS authors have the option to publish the peer review history of their article (what does this mean?). If published, this will include your full peer review and any attached files.

Reviewer #1: No

Reviewer #2: **Yes: **Javier S. Tellechea

---

## [Author Response · Author response to Decision Letter 0]

20 Oct 2023

Dear Editor, dear Reviewers, 

Thank you for your very helpful feedback. All comments have been taken into account, carefully considered and incorporated into the new version of the paper or discussed in the text given in the "rebuttal letter" in detail. In that document, our comments are written in bold italics.

Sincerely, Lara S. Burchardt on behalf of all co-authors

---

## [Editor Report · Decision Letter 1]

23 Nov 2023

Rhythmic properties of Sciaena umbra calls across space and time in the Mediterranean Sea

PONE-D-23-21637R1

Dear Dr. Lara S. Burchardt,

We’re pleased to inform you that your manuscript has been judged scientifically suitable for publication and will be formally accepted for publication once it meets all outstanding technical requirements.

Kind regards,

Javier S. Tellechea

Guest Editor

PLOS ONE

Additional Editor Comments (optional):

Dear authors,

based on the latest version sent by you, I think that the manuscript meets the reviewers' requirements, and is ready to be accepted for publication in PLOS ONE. Furthermore, I also think that the reviewers agreed on several aspects and above all that the work is pioneering on the subject.

I think that the reviewers' recommendations were met, and that the manuscript, in my opinion, now are much more robust, and shows the quality of its research. Therefore, I am pleased to tell you that I have given my suggestion for your manuscript to be published in PLOS ONE.

Greetings

Dr. Javier S. Tellechea

Ultrasonic Acoustics Laboratory, Institute of Physics, University of the Republic, Uruguay.

---

## [Editor Report · Acceptance letter]

30 Jan 2024

PONE-D-23-21637R1 

PLOS ONE

Dear Dr. Burchardt, 

I'm pleased to inform you that your manuscript has been deemed suitable for publication in PLOS ONE. Congratulations! Your manuscript is now being handed over to our production team.

Kind regards, 

on behalf of

Dr. Javier Tellechea 

Guest Editor

PLOS ONE